# Decreased virtual water out-flows from the Yellow River Basin are increasingly critical to China

Shuang Song[a, b], Shuai Wang[a, b], Xutong Wu[a, b], Yongyuan Huang[c], and Bojie Fu[a, b]

[a] State Key Laboratory of Earth Surface Processes and Resource Ecology, Faculty of Geographical Science, Beijing Normal University, Beijing 100875, P.R. China
[b] Institute of Land Surface System and Sustainability, Faculty of Geographical Science, Beijing Normal University, Beijing 100875, P.R. China
[c] College of Urban and Environmental Sciences, Peking University, Beijing 100871, P.R. China

**Correspondence:** Shuai Wang (shuaiwang@bnu.edu.cn)

**Abstract.** Water scarcity is an emerging threat to food security and socio-economic prosperity, and it is crucial to assess crop production response to water scarcity in large river basins. The water footprint, which considers water use in supply chains, provides a powerful tool for assessing the contributions of water resources within a certain region by tracking the volume and structure of virtual water flows. In this study of the structure of the water footprint network from a complexity perspective, we reassessed the significance of water resources for crop services in a large river basin with a severe water shortage -the Yellow River Basin (YRB) of China. The temporal increase of the complexity index indicated that the Virtual Water out-Flows (VWF) from the YRB were becoming increasingly critical to China; i.e., the ability of YRB to produce crops boosted the difficulty of its water being replaced by water exporting from other basins. Decomposition of complexity suggested that during the 1980s to 2000s, the temporally increased complexity was due mainly to the lack of competitors and the increasing uniqueness of crops supporting VWF. This complexity deeply embedded the YRB into the footprints of a water network that facilitated further development with constrained water resources. Still, it also reinforced reliance from other regions on YRB's scarce water. Based on this analysis, we suggest that resource regulation should be carried out appropriately to ensure ecological sustainability and high-quality development of river basins.

## 1 Introduction

Water scarcity is an emerging threat to food security and socio-economic prosperity (Zhou et al., 2020; Liu et al., 2017; Dolan et al., 2021; Mekonnen and Hoekstra, 2016). Water resources play an essential role in producing crops within a river basin. The export of those crops to other river basins can greatly exacerbate the negative consequences of water shortages. Although China's per capita water resources are only 25% of the global average, China is trying to decrease its dependence on the imported crop, become self-sufficient in food production, and ensure that its food supply is secure. A reasonable assessment of the contribution of the water resources in a river basin to domestic crops supplies will be the first step in balancing the water-food nexus (Wang et al., 2020). The Yellow River Basin (YRB) is China's most critical agricultural production area. Although the YRB uses only 2.66% of the water resources in China, the provinces it flows through account for over 35.63%

of the national grain production (data from official statistics, https://data.stats.gov.cn, last access: March 21, 2022), with $41\%$ of its grain production consumed outside the basin (Zhuo et al., 2016b, 2020). Because water consumption for agricultural purposes once accounted for 80% of the natural runoff of water in the YRB and caused the Yellow River to dry, the supply of water in the YRB is now strictly controlled (Wang et al., 2019). Although the availability of water resources constrains agricultural production in the YRB, the national crop yield per unit of water resources has been increasing rapidly (Zhou et al., 2020). An assessment of the changing status of agricultural production in the YRB may therefore provide a valuable example to guide the development of water resources in other resource-deficient basins.

The water footprint and virtual water (Conference on Priorities for Water Resources Allocation and Management, 1993), a geographically explicit indicator that involves water use in supply chains, has provided a powerful tool for assessing the contribution of water resources to a basin and tracking the transfer of water resources across regions (Jaramillo and Destouni, 2015; Oki and Kanae, 2004). Virtual water can be associated with specific products that transfer through complex trade relationships between geographic units (Hoekstra, 2014). Virtual water trade networks consist of Virtual Water out-Flows (VWF) that are embedded in production and consumption trajectories of crops (Oki and Kanae, 2004; Chini et al., 2018; Bae and Dall'erba, 2018). For example, the VWF from China's water-rich south to its water-scarce north has been quantified, and the network represented by crop trade between provinces has been mapped (Zhai et al., 2019; Zhuo et al., 2016a). However, because these volume-based studies have ignored the complexity of crop supply and demands (linked with trade) and the uniqueness of each basin (determined by regional characteristics), they have failed to consider the structure of water footprint networks. The structure of a water footprint network reflects the inherent heterogeneity of the distribution of the resources and the pattern of production and consumption. This heterogeneity is consistent with the fact that a water resource cannot be simply replaced by the same volume of water in another basin (Yu and Ding, 2021; Zhuo et al., 2016a; Li et al., 2020). This replacement problem reflects that water is not the only resource needed for widely circulated agricultural products. Because of path dependence, others resources, such as the unique hydrothermal conditions and the status of infrastructure within a basin, all determine the position of the basin (or a region) in a water footprint network (Best, 2019). Complexity, a rapidly developing concept in fields of study such as complexity science, network science, and development economics, represents the "capacity" that is embedded in a certain region, based on measurement of its position in structural terms (Hidalgo, 2021; Arthur, 2021; Meng et al., 2020; Hidalgo et al., 2007). For water resources that are unevenly distributed geographically, complexity can characterize a basin's overall capacity to provide water services required for national crop production. Because complexity-based metrics have been extensively studied in the empirical assessment of economic vitality through a bipartite-networks approach, the concept of complexity should be taken into consideration in the context of water footprint networks to upgrade the toolbox used for integrated water resources management (Hidalgo, 2021; Hidalgo and Hausmann, 2009; Liu et al., 2017). With such a toolbox, the regional significance of the water supply services provided by the basin through agricultural production can be comprehensively reassessed in a way that takes into account structural factors (Figure 1).

Recently, the growing idea is that the heart of China's crop production should cease to be the water-poor basin (e.g., the YRB) and that other water-rich basin (southern areas of China) should contribute more (Liu et al., 2021; Zhuo et al., 2016a). Another call is to ease the water shortage in the YRB by transferring water across the basins as soon as possible (Liu et al., 2021). These

strategies rely on a comprehensive assessment of the significance of water resources. Thus it would be an oversight to ignore the water footprint network in a structural context because each basin differs in terms of its capacity to use water resources to provide services for crop production (Li et al., 2020; Mekonnen and Hoekstra, 2020, 2011). In this study, we took a complexity perspective concerning water footprint networks and assessed the significance of the YRB to China's crop supply. Our results showed that although the YRB had reduced its virtual water out-flow because of resource constraints, its importance to crop production in China increased when considering water footprint networks. The complexity of the water footprint network enabled our approach to provide a new perspective for understanding the changes in the status of a basin with a severe water shortage concerning national crop production.

## 2 Methods

An overall method route illustrated here (Figure 1):

### 2.1 Dataset and VWF estimation

We produced provinces-crops associations by using a rich dataset on the water footprint of crop production and consumption in China, which is openly accessible on (waterfootprint.org). The dataset contains annual statistics for the period 1978-2008 for 22 individual crops (wheat, maize, rice, sorghum, barley, millet, potato, sweet potato, soybean, groundnuts, sunflower, rapeseed, sugar beet, sugar cane, cotton, spinach, tomato, cabbage, apple, grapes, tea, tobacco). For each crop, The annual water footprint of both production and consumption are available at the province level of China ($n = 31$, no data in Taiwan, Hong Kong, and Macau) are available. The dataset was widely used when evaluating Chinese virtual water flows (Xie et al., 2020; Sun et al., 2021; Zhuo et al., 2016b). As national water footprint in total production and total consumption is equal for each crop $j$, the difference between the water footprint of production and consumption indicates the surplus partial for province $i$, considered to be Virtual Water out-Flows (VFW):

$$
\begin{cases}
D_{ij} = F_{i,j}^{production} - F_{i,j}^{consuption}, & \text{if } F_{i,j}^{production} - F_{i,j}^{consuption} > 0 \\
D_{ij} = 0, & \text{if } F_{i,j}^{production} - F_{i,j}^{consuption} \leq 0
\end{cases}
\tag{1}
$$

In this way, we obtained the annual volume matrix of VFW $D_{ij}$ in year $y$, where $i$ and $j$ indicate a specific province and a particular crop, $F_{production}$ and $F_{consumption}$ are virtual water embedded in crops production and consumption directly from the original dataset. We focus on outflows, so virtual water inputs (i.e., if consumption is more extensive than its production) are not considered (set to zero).

### 2.2 Construction of network

To construct a bipartite network, we reduce the VWF volume matrix $D_{ij}^{y}$ to a binary matrix to indicate whether there is a linking edge between a province and a crop (for capturing the network topology). The Relative Comparative Advantage (RCA) procedure was used to construct the bipartite network between provinces and crops. The RCA of production referred to a

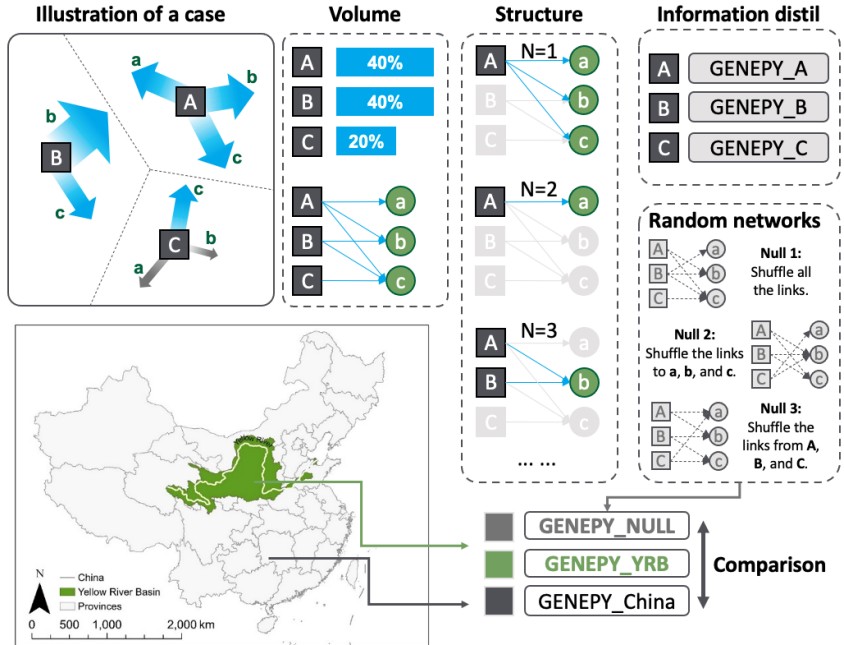

**Figure 1.** Virtual water transfers between regions through multiple dominant crops. When the proportion of VWF from a region is large enough when considering a specific crop, we establish a connection between the region and the crop. In that way, we abstract region-crop bipartite networks for analysis. Here, we give a more straightforward illustration of our method: (1) Illustrating a nation comprised of three regions ($A$, $B$, and $C$) where three types of crops ($a$, $b$, and $c$) can be the productions for supporting virtual water flows (VWF). Region $B$ doesn't produce crop $a$, and region $C$ produce a negligible crop $a$ and $b$. (2) When considering VWF volume, it is hard to compare regions $A$ and $B$, as they support 40% national total VWF. (3) However, when the structure is involved, their position in region-crop bipartite networks are different, typically in their differences of diversification ($N = 1$), uniqueness ($N = 2$) and competitiveness ($N = 3$) (the parameter $N$ refers to levels of decomposition, detailed interpretation in section 2.4). (4) Therefore, considering both volume (by ignoring eligible VWF crops) and structure, the $GENEPY$ index is a method for distilling information (see section 2.3). Our main results derive from comparisons between indexes of YRB, China, and random networks (as a benchmark). We generated three different null models by shuffling links in different ways and Table 2 gives them mathematical descriptions.

region's (provinces in this study) export of a particular product (crops, here) in terms of its proportion of the total trade of that product. An analogy can be made to the VWF ($D_{ij}$) of province $i$ as a proportion of the national total volume of VWF ($\sum_i D_{ij}$), embedded in a specific crop $j$. A higher RCA indicates a larger part of shares among the total national VWF, i.e., more virtual water exported from the province through a specific crop. Firstly, for a specific year $y$, $RCA$ matrix of province $i$ and crop $j$ are calculated according to (Balassa, 1965; Dolan et al., 2021; Sciarra et al., 2020):

$$RCA_{ij} = \frac{D_{ij}/\sum_j D_{ij}}{\sum_i D_{ij}/\sum_{ij} D_{ij}} \tag{2}$$

Then, for $y \in [1978, 2008]$, we constructed such bipartite network $RCA_{ij}$ year by year to capture topology changes of Chinese VWF by time. Then, the network matrix $M$ is given by $M_{ij} = 1$ if $RCA_{ij} \geq 1$, and 0 if $RCA_{ij} < 1$.

## 2.3 Quantitative metrics of complexity

We used a GENEPY index to distil information on the networks in reference to economic complexity (Sciarra et al., 2020):

$$GENEPY_i = \left( \sum_{x=1}^{2} \lambda_x X_{i,x}^2 \right)^2 + 2 \sum_{x=1}^{2} \lambda_x^2 X_{i,x}^2 \tag{3}$$

where $X_{i,1}$ and $X_{i,2}$ are the normalized eigenvectors of province $i$ corresponding to the first two largest eigenvalues $\lambda_1$ and $\lambda_2$ of the proximity matrix $N_{ii^*}$:

$$\begin{cases} N_{ii^*} = \sum_j \dfrac{M_{ij} M_{i^* j}}{k_i k_{i^*} \left( k'_j \right)^2}, & \text{if } i \neq i^* \text{ where, } k_i = \sum_j M_{ij}, k'_j = \sum_i M_{ij}/k_i \\ N_{ii^*} = 0, & \text{if } i = i^* \end{cases} \tag{4}$$

Here, $M$ is the constructed network matrix, $k_i$ is the degree (how many types of crops connected in the network) of the province $i$ and $k'_j$ represents the degree of a crop corrected by how easy it is found within the network. The redundant information of the self-proximity (i.e., when $i = i^*$) is deleted by setting corresponding values to zero. In addition, the symmetric square matrix $N$ is interpreted as the mathematical description of the weighted topology of the network -such that the provinces are the nodes and the similarities between the VWF-supporting crops are the links connecting them. Then, the eigenvector centrality of the nodes (i.e., the matrix $N$) can be a helpful tool to interpret the complexity of the network. In a practical sense, two eigenvalues vectors $X_{i,1}$ and $X_{i,2}$ of each province $i$ can be combined into unique metrics to distil its ability in producing highly competitive crops. A higher GENEPY index indicates a potential superiority (VWF embedded are more accessible and irreplaceable) in the current VWF networks. For more details in math, we refer the reader to Sciarra et al. (2020). In this way, based on the idea of dimensionality reduction, we could use the average GENEPY index of the YRB to simply assess its importance to the virtual water of China:

$$\begin{cases} GENEPY_{YRB} = \frac{1}{n} * \sum_{i \in YRB}^{n} (GENEPY_i) \\ GENEPY_{China} = \frac{1}{N} * \sum_{i \in China}^{N} (GENEPY_i) \end{cases} \tag{5}$$

Here, $i \in YRB$ indicates that $i$ is one of the significant provinces that heavily rely on water resources in the YRB, $GENEPY_i$ is the complexity index of the province $i$ according to the **equation (3)**.

## 2.4 Decomposition of complexity

Decomposition where "Reflection" applied, of the complexity index "GENEPY" explains the main factors of network changes. The Reflection Method can describe a structure of bipartite network (Hidalgo and Hausmann, 2009; Hidalgo, 2021). To a certain province $i$, number of existed connections $k_{i,N}$ are vary according to different reflecting times $N$:

$$\begin{cases} k_{i,N} = \frac{1}{k_{i,0}} \sum_j M_{ij} k_{j,N-1}, \\ k_{j,N} = \frac{1}{k_{j,0}} \sum_i M_{ij} k_{i,N-1} \end{cases} \tag{6}$$

where $j$ represents a certain crop and $i$ a certain province, $M_{ij}$ defines the network, and $k_{i,0}$ represents the observed levels of diversification of a province (the number of products exported by that province). When reflecting times $N$ takes different integer values, it adds further potential connections to the total number in the previous layer (i.e., $N-1$) of the network. We therefore characterized each province $p$ through the vector $\boldsymbol{k_i}(\boldsymbol{k_{i,1}}, \boldsymbol{k_{i,2}}, \boldsymbol{k_{i,3}})$ in its different dimensions. For example, $N=1$, with initial conditions given by the degree (i.e., the number of links of provinces and crops): $k_{i,0} = k_i$, same as $k_i = \sum_j M_{ij}$ in **equation 4**.

**Table 1.** Interpretations of the first three pairs of variables describing the province-crop network through the method of reflections.

|  | Definition | Working name | Description |
|---|---|---|---|
| N=1 | $k_{i,1}$ | Diversification | How many products are exported by province $i$? |
| N=2 | $k_{i,2}$ | Uniqueness | How common are the crops exported by province $i$? |
| N=3 | $k_{i,3}$ | Competitiveness | How diversified are provinces exporting crops similar to those of province $i$? |

With reference to existing complexity studies, the first three major dimensions can therefore be intuitively explained when $N=1$, $N=2$ and $N=3$ as summarized in Table 1. When the reflection method is used in this way, it reflects the crop diversification, crop uniqueness and regional competitiveness of the water footprint out-flow of the YRB. Although we could have used the reflection approach to continue iterating for more complex explanations, decomposing complexity into three steps helped explain the changes in complexity more clearly and intuitively.

## 2.5 Null models and sensitivity tests

We randomly created provincial-crop bipartite networks for a sensitivity test. We calculated the same metrics as a comparable reference value to decide whether the influence of networks structure was trivial (Figure 1). The idea behind the randomization procedure is that we can create a null model starting from the data but shuffling the links of the network while conserving some of its statistical properties. We randomly generated (executed by Python 3.9 and Numpy 1.2) three scenarios where the bipartite networks have the same (1) number of edges, (2) edge sequences on provinces, and (3) edge sequences on crops, with the original dataset, respectively (Table 2 and Figure 1).

**Table 2.** How different null models were generated.

|  | Null model 1 | Null model 2 | Null model 3 |
|---|---|---|---|
| Number of links | $=M_{ij}$ | $=M_{ij}$ | $=M_{ij}$ |
| $k_{i,0}$ sequence | $\neq M_{ij}$ | $\neq M_{ij}$ | $=M_{ij}$ |
| $k_{j,0}$ sequence | $\neq M_{ij}$ | $=M_{ij}$ | $\neq M_{ij}$ |

## 3  Results

### 3.1  Increasing complexity with decreasing VWF volume

The total volume of virtual water out-flows from the provinces in the YRB decreased continuously during the study period (Figure 2 A), and its proportion of the national total decreased by an even more significant percentage (Figure 2 B). In 1978, there were five provinces in the YRB whose virtual water out-flows (VWF) exceeded the national average, but there were only three of them in 2008, and their overall ranking had decreased significantly. Though the total volume, share, and ranking of VWF across provinces of the YRB were all decreasing, the average complexity index of the YRB was holistically higher than

that of China (Figure 3 A). The gap between the two increased rapidly after 1985. It reached its widest point in 1993 when the average complexity index of the YRB was about 1.4 times the national average (Figure 3 B). After then, the difference between the two decreased with some fluctuations, but the complexity of the YRB remained about 1.2 times the national average (Figure 3 B).

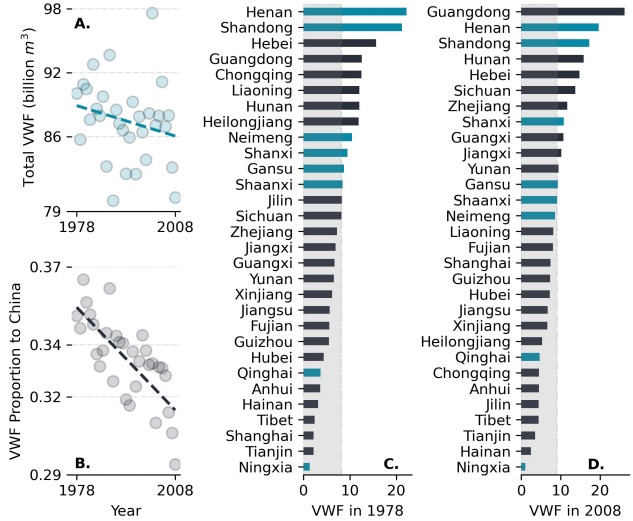

**Figure 2.** Virtual Water out-Flows (VWF) of the Yellow River Basin (YRB) changes from 1978 to 2008. **A.** Total VWF in the YRB and China. **B.** Proportion of total VWF in the YRB to the national volume. **C.** Ranking of VWF in Chinese provinces in 1978. Blue bars are provinces in the YRB. **D.** Ranking of VWF in Chinese provinces in 2008. Blue bars are provinces in the YRB.

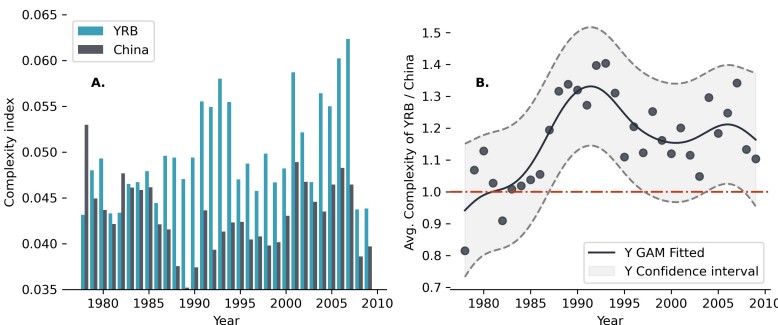

**Figure 3. A.** Average complexity index of YRB and of China from 1978 to 2008. **B.** Ratios of average complexity index of YRB to that of China from 1978 to 2008. The solid line was fitted with Generalized Additive Models (GAMs). The Gray shaded area indicates a 90% confidence interval. The red dashed line indicates the baseline where the average complexity index of the YRB is equal to that of China.

## 3.2 Decomposition of changing complexity

Firstly, an indication of crop diversification ($N = 1$, Figure 4A) was that almost every region was transferring virtual water through 7-8 dominant crops on average. The peak of diversification occurred before about 2000 when the major crop species supporting virtual water transfer were most abundant. Still, there is no significant difference between the YRB, China, and the random network (null models). When $N = 2$ (Figure 4B), however, both the YRB and the national average had a significantly ($p < 0.1$) lower number of competitors that transferred virtual water through their similar dominant crops than the random network. The implication is that the crops supporting VWF of the YRB were not all the same to other regions -and are usually more unique than other regions. The opposite nature of the trends of the YRB before the 2000s indicated that its uniqueness was rising and significantly higher (i.e., reflecting index was lower, $p < 0.1$) than the national average. For $N = 3$ (Figure 4C), whereas the national average of competitiveness was relatively similar to that of the random networks, the total number of types of crops that are competitors with the YRB were significantly lower between 1987 and 2000. The implication is that the competitiveness of the YRB exceeded the average level. In general, since the 1980s, the uniqueness and competitiveness of the YRB have been very different from that of the whole country. However, that difference gradually disappeared in the 2000s.

## 4 Discussion

The use of regional VWF, especially within a water footprint network, has been a common approach to assess the importance of water resources, but complex structures have not been comprehensively evaluated in these assessments (Mekonnen and Hoekstra, 2020, 2011; Fang et al., 2014). The Yellow River, an agriculture-oriented basin with substantial heterogeneity in the upper, middle, and lower reaches, has scarce water resources that are the cornerstone of its crop production (Wang et al., 2019). With "The Reform and Opening" of China since 1978, domestic crop trades have gradually increased both in diversity and the volume, and that increase may have caused the total VWF volume to keep increasing. At the same time, however, the

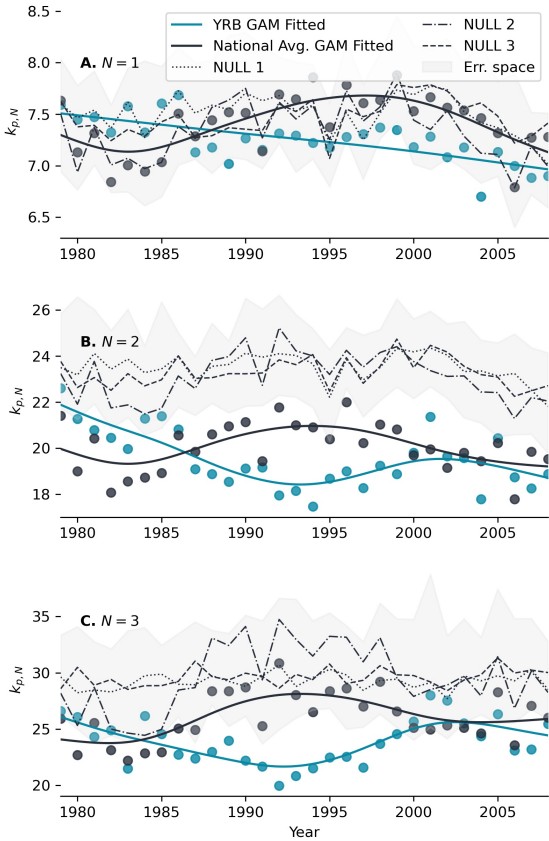

**Figure 4.** Trends of different dimensions based on the reflecting method decomposed from complexity (see the Methods *Sect. 2.3 and Table 1*). Blue and grey colours show the average levels of the YRB and China, respectively. **A.** $N = 1$, number of major crops that support virtual water out-flow (VWF), an indication of diversification of a certain region. **B.** $N = 2$, the total number of provinces where similar dominant crops supporting VWF are included, a smaller number indicating the greater uniqueness of the particular region. **C.** $N = 3$, the total number of dominant crop types supporting VWF from all competitors (who have similar dominant crops). A smaller number of competitors means higher competitiveness of the YRB.

YRB was the first basin to apply a water allocation scheme because extreme water shortages caused the river to frequently
dry after the 1970s (Wang et al., 2019). Therefore, during the time that the national VWF was transferred mainly from the north (where the YRB is located) to the south (Zhuo et al., 2016b), the contribution of the YRB, which is deficient in water resources, decreased in volume (Figure 2). However, considering the structure of the crop-water footprint bipartite network,

that did not indicate the YRB had been decreasing in importance to China. On the contrary, our complexity-based analysis revealed the differences in complexity between the YRB and China since 1978. As a result, the YRB's water footprints have been complex for other basins to replace crop production. Our decomposition results suggested that this difficulty was due mainly to a low number of competitors and the increasing uniqueness of crops supporting VWF from the YRB. The Yellow River is a large river that crosses an arid, semi-arid, and monsoon climate zone. The small number of competitors and the relatively high uniqueness of the YRB depend partially on the unique geographical conditions in the YRB (Fu et al., 2017). For example, high-quality apples produced in the fertile but arid Loess Plateau in the middle reaches of the YRB and the wet areas in the lower reaches together account for more than 90% of the total apple production in China regions can compete with the YRB. The increasing uniqueness of the YRB means that its ability to improve the quality of its agricultural products and push competitors out of the crop supply network is growing. Those regions that can compete with a particular crop from the YRB (such as Xinjiang, which also exports many high-quality apples) are limited by an extreme shortage of water resources and cannot increase diversification, hence lacking overall competitiveness.

Assessing the value of natural resources has been a constant problem because of the trade-off between economic efficiency, social equity and resource availability (Dalin et al., 2015; Grafton et al., 2018; Yoon et al., 2021). The "Porter hypothesis" of the environment has proposed that environmental regulation can stimulate technological innovation. Similarly, the stimulus of resource scarcity can improve optimization of a market allocation for efficient use of resource (Wagner, 2004; Luptáčik, 2010). The complexity of the YRB had risen above the national average since about 1987, when the river basin was regulated. Because of water scarcity, the Chinese government required provinces to adhere to strict resource quotas (Wang et al., 2018). During this period, although the total VWF of the YRB and its proportion decreased, the increasing complexity indicated that the resource-constrained YRB was gaining market advantages. In different ways, this advantage has manifested itself as an increase of the position of the YRB in the virtual water network and by an increase of both competitiveness and uniqueness (Figure 4) (Fang and Chen, 2015; Fang et al., 2014; Yang et al., 2012). After the 1990s, the complexity index of China evidenced a similar trend of improvement, which was concomitant with the period when the strict control of water resources and a transformation to water-conservation were implemented throughout the whole country (Zhou et al., 2020; Liu and Yang, 2012). According to the literature (Zhou et al., 2020), the slowdown of the growth of China's water consumption can be divided into two stages by the 1990s. While growth during the earlier period occurred mainly in arid and semi-arid areas (e.g., the YRB), growth during the latter period affected the whole country. Therefore, the increase of the national average level of complexity lagged behind that of the YRB. This lag was probably the result of the rest of China pursuing structural advantages later than the YRB. Traditional development economic theory points out that the use of resources with comparative advantage is a prerequisite for producing an economically efficient division of labour and a marketing network (Hidalgo and Hausmann, 2009). In this case, however, before water resources are severely restricted, it may not be more cost-effective to intensify a comparative advantage than to expand resource investment. This conclusion is consistent with the theories related to the "peak water use" and the "efficiency paradox." (Gleick and Palaniappan, 2010; Grafton et al., 2018). Thus, the complexity index may depict the process by which the YRB and then China pursued this structural advantage within the water footprint network as constrained by resources.

In gradually being deeply embedded in the water footprint network, the quality of crops in the YRB has constantly been improved. That improvement has become the structural driving force behind basin development. According to our analysis, the Yellow River Basin (YRB) has already developed competitiveness. As a result, the VWF of the YRB was well-structured into national Networks. Therefore, it is not easy to replace crop productions with other regions. As a result, water diverted to the Yellow River Basin is an option to balance water shortage and crop production with food security (Long et al., 2020). However, it must be pointed out that the YRB is a basin with a severe water shortage, and the proposed water resource regulations were also promulgated during a crisis period when the river was drying up (Wang et al., 2019, 2018). Like a double-edged sword, the increased complexity facilitated the continued development of the YRB with constrained water resources and embedded it more deeply into the water footprints networks where scarce water could not be easily replaced. From this structural perspective, complexity provided a reminder to help guide China as quickly as possible from resource-dependent development to high-quality development that would enhance regional competitiveness. However, when there is a crisis in the availability of resources, regulations must be implemented. Those regulations may deepen structural issues and make it more challenging to solve the ecological crisis exposed completely.

Our research shows that introducing the concept of complexity into virtual water research is an exciting attempt with new perspectives, but there are still many limitations to this initial exploration. Complexity also depends on other factors in analyzed regions, such as policies, markets, infrastructure, labour resources. However, these are not the focus of this paper and are most similar within provinces. Moreover, no suitable water footprint data for an extended period to watershed boundaries are available is another challenge for focusing on water resources' influence on complexity. That is why we construct networks by regarding provinces as regional nodes. However, combinations between provincial (Zhuo et al., 2016b) and watershed datasets (Xie et al., 2020) can be better for accurate assessments. This study focuses more on changing trends rather than the mechanisms. Developing a more sophisticated dataset for further analysis can be one of the future directions. Furthermore, there is also a gap between VWF complexity and its influence on water resources management. If complexity (levels of a region embedded into VWF networks) is good or dangerous to water-poor and water-rich basins is still on an open discussion.

## 5 Conclusions

This paper took the structure of the water footprint network from a complexity perspective. It assessed the significance of water resources for crop services in a large river basin with a severe water shortage -the Yellow River Basin (YRB). From 1978 to 2008, the number of Virtual Water out-Flows (VWF) from the YRB and its percentage of the total VWF from the national total decreased significantly. The fact that the YRB has lagged behind the national VWF trend has probably related to restricted water use policies. However, our results showed that the complexity of the YRB increased and was significantly higher than the national average during the period from the 1980s to the 2000s. Decomposition of complexity suggested that this pattern was due mainly to few competitors and the increasing uniqueness of supporting VWF crops for the YRB. Based on an assessment of water conservation policies in China, we suggested that the initial promulgation of resource regulations was the key to a competitiveness-oriented transformation for crops production in the YRB. Subsequently, the increased complexity enabled the

240 YRB to develop with limited water resources. However, it also more deeply embedded the YRB into water footprints networks where scarce water cannot be easily replaced. From the complexity analysis, we point out that resource regulation should be carried out at an appropriate stage to ensure ecological sustainability and high-quality development of river basins.

*Code availability.* All code is open in my Github repository: SongshGeo/complexity-yrb

*Data availability.* Using published data, available from waterfootprint.org

*Author contributions.* Shuai Wang and Bojie Fu designed this research, Shuang Song performed the research and analyzed data, Shuang Song and Yongyuan Huang wrote the paper, Xutong Wu and Yongyuan Huang revised, polished the manuscript, and gave major advice.

*Competing interests.* The authors declare no competing interests.

*Acknowledgements.* Funding was provided by the National Natural Science Foundation of China (CN) (Grant Nos. NSFC 42041007). We also thank Yinan Xiao (School of Artificial Intelligence, Peking University) verified our application of the mathematical method.

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
