# Peer review of "Decreased virtual water out-flows from the Yellow River Basin are increasingly critical to China"

_Hydrology and Earth System Sciences, 2021_

## Author Response (AR1)

**Response letter:**

Dear Editor in Chief,

Thank you very much and the two anonymous reviewers for your valuable comments on our work. We have amended mistakes, enriched the literature, addressed flaws, and improved the description of our research. Following is the point-to-point responses to the comments from two reviewers, a demo code to show our correct adaptation of the original algorithm included in the supplementary materials. We hope this improved manuscript will satisfy you.

Thank you very much for your further attention and consideration. We are looking forward to hearing from you soon.

Best Regards

Yours Sincerely,

Shuang Song & Shuai Wang (on behalf of the author's team)

State Key Laboratory of Earth Surface Processes and Resource Ecology,

Faculty of Geographical Science,

Beijing Normal University

Beijing 100875, China

Tel: +86-10-58806579

Fax: +86-10-58806579

Email: songshgeo@mail.bnu.edu.cn

**Comments Response to reviewer 1**

**Overall comments**

*Based on the concept of "complexity" in complex economics, this manuscript points out that the virtual water flow for trade in the Yellow River basin is decreasing, but its irreplaceability is increasing. There are significant flaws in literature citation and method description, so it is hard for me to recommend the current version of the manuscript to be published in the HESS. Following are some comments:*

**Response**:

Thanks for taking the time to comment on our manuscript. We are very sorry that the manuscript was confusing to you. We spent more effort sorting out the method section and hope it helps readers follow our work by our revised method section. Furthermore, we added some more citations to support our analysis when necessary, in the revised manuscript.

**Detailed comments**

*1. The methodological part of the manuscript is not clear and accurate. In the process of model construction, there are obvious errors in all corner labels of the author, such as . According to Sciarra et al. (2020), the author wrongly defined the meanings of and in the calculation in this formula, especially when using the same qualification conditions as in Sciarra et al. (2020). The same error occurs in Equation 3. In addition, the content of Formula 5 and 6 is the same, and the author does not give an effective description of its essential difference. Therefore, I doubt the rationality of the calculation method in this manuscript, and the results revealed are debatable.*

**Response**: Thank you for your suggestion, and we are sorry for your confusion about the methodological part. In the previous version, unfortunately, while we directly referred to the algorithm developed by *Sciarra*, we have to apply it in another analysis scale. This is because the original corner labels were opposite to ours: -in *Sciarra*'s article, they use abbreviations (country, $c$; production, $p$) to indicate regions and production; In our previous version, however, we used opposite abbreviations (province, $p$; crop, $c$) as indications of regions and productions, respectively. It is indeed very counterintuitive, so I've completely changed them to $i$ and $j$ (see the method introduction in attachments). We hope this helps reduce your confusion when tracing the methods:

*[P4, L90-L91]* Firstly, for a specific year $y$, $RCA$ matrix of province $i$ and crop $j$ are calculated according to:

$$RCA_{ij} = \frac{D_{ij}/\sum_j D_{ij}}{\sum_i D_{ij}/\sum_{ij} D_{ij}}$$

Then, for $y \in [1978,2008]$, we constructed such bipartite network $RCA_{ij}$ year by year to capture topology changes of Chinese VWF by time. Then, the network matrix $M$ is given by $M_{ij} = 1$ if $RCA_{ij} \geq 1$, and 0 if $RCA_{ij} < 1$.

*[P5, L96-L103]* We used a GENEPY index to distil information on the networks in reference to economic complexity (Sciarra et al., 2020):

$$GENEPY_i = \left(\sum_{x=1}^{2} \lambda_x X_{i,x}^2\right)^2 + 2\sum_{x=1}^{2} \lambda_x^2 X_{i,x}^2$$

where $X_{i,1}$ and $X_{i,2}$ are the normalized eigenvectors of province $i$ corresponding to the first two largest eigenvalues $\lambda_1$ and $\lambda_2$ of the proximity matrix $N_{ii^*}$:

$$\begin{cases} N_{ii^*} = \sum_j \frac{M_{ij}M_{i^*j}}{k_i k_{i^*} \left(k_j'\right)^2}, & \text{if } i \neq i^* \text{ where, } k_i = \sum_j M_{ij}, k_j' = \sum_i M_{ij}/k_i \\ N_{ii^*} = 0, & \text{if } i = i^* \end{cases}$$

Here, $M$ is the constructed network matrix, $k_i$ is the degree (how many types of crops connected in the network) of the province $i$ and $k_j'$ represents the degree of a crop corrected by how easy it is found within the network. The redundant information of the self-proximity (i.e., when $i = i^*$) is deleted by setting corresponding values to zero.

Finally, we confirm that the algorithm worked correctly in programming. We rechecked that the algorithm codes were correct, and attach a demo code so that you can revalidate our calculation of the *GENEPY index* proposed by *Sciarra*. At the same time, all our source codes are openly accessible on GitHub (https://github.com/SongshGeo/complexity_yrb).

*2. There is little information on how the VWF are simulated.*

**Response**:

Thanks for your comment, and we now make this much more apparent in a new subsection **Dataset and VWF estimation**:

*[P3, L75-L82]* As national water footprint in total production and total consumption is equal for each crop $j$, the difference between the water footprint of production and

consumption indicates the surplus partial for province $i$, considered to be Virtual Water outflows (VFW):

$$D_{ij} = F_{i,j}^{production} - F_{i,j}^{consuption}, \text{if } F_{i,j}^{production} - F_{i,j}^{consuption} > 0$$

$$D_{ij} = 0, \text{if } F_{i,j}^{production} - F_{i,j}^{consuption} \geq 0$$

In this way, we obtained the annual volume matrix of VFW $D_{ij}$ in year $y$, where $i$ and $j$ indicate a specific province and a particular crop, $F_{production}$ and $F_{consumption}$ are virtual water embedded in crops production and consumption directly from the original dataset. We focus on outflows, so virtual water inputs (i.e., if consumption is more extensive than its production) are not considered (set to zero).

*3. Information on data sources is lacking!*

**Response**:

Thanks for your comment, and now we make this much more explicit in a new subsection **Dataset and VWF estimation**:

[***P3, L69-L75***] We produced provinces-crops associations by using a rich dataset on the water footprint of crop production and consumption in China, which is openly accessible on (https://waterfootprint.org/en/resources/waterstat/wf-crop-production-and-consumption-china/). The dataset contains annual statistics for the period 1978-2008 for 22 individual crops (wheat, maize, rice, sorghum, barley, millet, potato, sweet potato, soybean, groundnuts, sunflower, rapeseed, sugar beet, sugar cane, cotton, spinach, tomato, cabbage, apple, grapes, tea, tobacco). For each crop, The annual water footprint of both production and consumption are available at the province level of China ($n = 31$, no data in Taiwan, Hong Kong, and Macau) are available. The dataset was widely used when evaluating Chinese virtual water flows (Xie et al., 2020; Sun et al., 2021; Zhuo et al., 2016b).

*4. In general, the boundaries of river basins are not equivalent to the administrative boundaries. Since this paper focuses on the Yellow River basin, the quantification results at the provincial level cannot accurately reflect the situation of the basin itself. Or it should be clearly described that how the part of each province within the basin was identified.*

**Response**:

Thanks for the comment. The mismatch between administrative and hydrological areas are a widespread problem, and here are reasons why this problem does not damage our main goal:

First and foremost, we obtained core conclusions by comparing provincial averages in the Yellow River Basin (YRB) and the national average levels, indicating that changing trend is more important than the complexity index value in this study. Furthermore, the national average includes provinces that depend heavily on the Yellow River (Qinghai, Gansu, Ningxia, Neimeng, Shaanxi, Shanxi, Henan, and Shandong Province). Therefore, when comparing the two mean values, the differences in their trends can show that the relevant provinces are different as a whole.

Secondly, as we mentioned in the article, complexity also depends on other factors in the region, such as policies, markets, infrastructure, human resources, etc. These variables, which are not the focus of this paper, are most similar within provinces. Therefore, although this paper focuses on the "complexity" of basin-scale and water resources as the main object of analysis, the network construction with the province as the node is just the proper scale that can ignore these factors.

Finally, no suitable footprint data corresponding to watershed boundaries are available. Suppose the virtual water data at the basin scale can be more accurately combined with social and economic data at the administrative hierarchy. In that case, it will be a more detailed quantitative analysis. However, the water footprint of crops is currently difficult to obtain data at more minor spatial scales over such a long period. This study focuses more on change trends analysis, so it does not develop a more sophisticated data set that combines watershed with administrative scale. We have now mentioned this limitation in the discussion so that future researchers using this method can consider further studies on a more detailed scale:

> [*P11, L218-L227*] Our research shows that introducing the concept of complexity into virtual water research is an exciting attempt with new perspectives, but there are still many limitations to this initial exploration. Complexity also depends on other factors in analysed regions, such as policies, markets, infrastructure, labour resources, etc. These are not the focus of this paper and are most similar within provinces. Moreover, no suitable water footprint data for an extended period to watershed boundaries are available is another challenge for focusing on water resources' influence on complexity. That's why we construct networks by regarding provinces as regional nodes. However, combinations between provincial (*3*) and watershed datasets (*1*) can be better for accurate assessments. This study focuses more on changing trends rather than the mechanisms. Developing a more sophisticated dataset for further analysis can be one of the future directions. Furthermore, there is also a gap between VWF complexity and its influence on water

resources management. If complexity (levels of a region embedded into VWF networks) is good or dangerous to water-poor and water-rich basins is still on an open discussion.

*5. L21-29: For descriptions of the current status of water resources in the Yellow River basin, it is important to cite official statistics rather than secondary literature.*

**Response**:

Thanks for the comment, and you are right. Now we calculated the data again according to official statistics:

> [***P1, L21-L24***] The Yellow River Basin (YRB) is China's most critical agricultural production area. Although the YRB uses only 2.66% of the water resources in China, the provinces it flows through account for over 35.63% of the national grain production (data from official statistics, https://data.stats.gov.cn, last access: 16, Feb 2022), with 41% of its grain production consumed outside the basin.

*6. L33-34: As for the derivation of the concept of water footprint and virtual water, the literature cited here is not clear and unrepresentative.*

**Response**:

Thanks a lot for pointing this out. We may add references to some of the original and most representative literature to trace the birth and development of the concept in the mentioned paragraph:

> *[P2, L30-L34]* The water footprint and virtual water (Conference on Priorities for Water Resources Allocation and Management, 1993), a geographically explicit indicator that involves water use in supply chains, has provided a powerful tool for assessing the contribution of water resources to a basin and tracking the transfer of water resources across regions (Jaramillo and Destouni, 2015; Oki and Kanae. 2004). Virtual water can be associated with specific products that transfer through complex trade relationships.

*7. L56-57: Reference citation format error.*

**Response**: Thanks a lot for your circumspection. We rectified this mistake.

*8. In Figure 2 c, why choose to show 1979 instead of 1978*

**Response**: Thanks a lot for pointing this out. That was a mistyping, and we rectified it now.

*9. L154: Spelling mistake: production.*

**Response**: Thanks for your circumspection, and we rectified this misspelling now.

**Bibliography**

P. Xie, L. Zhuo, X. Yang, H. Huang, X. Gao, P. Wu, Spatial-temporal variations in blue and green water resources, water footprints and water scarcities in a large river basin: A case for the Yellow River basin. *Journal of Hydrology*. 590 (2020), doi:10.1016/j.jhydrol.2020.125222.

J. Sun, Y. Yin, S. Sun, Y. Wang, X. Yu, K. Yan, Review on research status of virtual water: The perspective of accounting methods, impact assessment and limitations. *Agricultural Water Management*. 243 (2021), doi:10.1016/j.agwat.2020.106407.

L. Zhuo, M. M. Mekonnen, A. Y. Hoekstra, Y. Wada, Inter- and intra-annual variation of water footprint of crops and blue water scarcity in the Yellow River basin (19612009). *Advances in Water Resources*. 87, 29–41 (2016).

Allan J A, 1993. Fortunately there are substitutes for water otherwise our hydro-political futures would be impossible. Proceedings of the Conference on Priorities for Water Resources Allocation and Management: Natural Resources and Engineering Advisers Conference, Southampton, pp. 13–26.

Oki T, Kanae S, 2004. Virtual water trade and water resources. *Water Science & Technology*, 49(7): 203-209.

Hoekstra A Y, 2014. Water scarcity challenges to business. *Nature Climate Change*, 4: 318-320.

**Comments Response to reviewer 2**

**Summary**

*The article explains how the complexity of virtual water footprint network determining the situation of crop production under water scarcity in the Yellow River Basin (YRB), China. The mechanism of increasing water footprint network complexity pushing the continuous development of the Yellow River Basin with limited water resources has been revealed and discussed. The authors emphasized the necessity of both ecological sustainability and high-quality development with the fact that water footprint network complexity can also increase of difficulty of relieving water stress by simply replacing scarce water.*

**General comment**

*The main conclusion of the article is supported by clear thinking and accurate data analysis. As the authors provide detailed interpretation of virtual water network related mechanisms over the YRB during historical period (1980s to 2000s), it is recommended to extend the time axis to more recent period and show projections for future scenarios of virtual water network and water scarcity or disasters within the YRB. The intuitiveness and readability of the Mathematical method description can be improved. A major revision is recommended for further works on this manuscript.*

**Response:** We appreciate the acknowledgement of our study. Now, a more understandable method section is available with improved text which I hope will satisfy you. We appreciate your proposal to extend the analysis. However, it is not easy to prolong our analysis up to now. Firstly, our core findings can be revealed now, by existing (as far as we know) the most relevant data. Secondly, producing a coherent dataset for prolonging the analysis or even predictions is not our primary goal. Finally, we have contacted the producer of that dataset, and their team is working with an update as it's meaningful, but a tough workload. Therefore, we kept our research design but much-improved description and interpretation of our manuscript. There will be a further explanation in the following detailed comments.

**Comment list**

*The detailed comments are listed below. 1. Easing water shortage in the YRB by transferring water across the basin may not contradict the main conclusion of this research. The complexity of water footprint network brings high uniqueness of the YRB and makes it not easily to be*

*replaced by other regions. Is this case, water transferring can be a useful approach at least for the current period. The authors may adjust the related sentences of the literature review here.*

**Response:** Thanks for the comment, and we cannot agree with the opinion more. Transferring water across the basin is an effective method to deal with the water shortage. Some more discussions have been added in a revised manuscript:

> [***P10 L206-L209***] According to our analysis, the Yellow River Basin (YRB) has already developed competitiveness. As a result, the VWF of the YRB was well-structured into national Networks. Therefore, it is not easy to replace crop productions with other regions. As a result, water diverted to the Yellow River Basin is an option to balance water shortage and crop production with food security (Long et a., 2020).

*2. Line 70: The research focuses on the historical evolution of virtual water footprint in the Yellow River Basin by applying data from 1980s to 2000s. Is there more recent dataset available especially for the previous decade?*

**Response:** Thanks for the comment. We considered an extension of the study period. However, an available dataset with enough crop types and virtual water is not supported (***a detailed introduction of the methods, including the datasets section, P3-P6***). We contacted the datasets' provider, but the latest update is still in production and unavailable. Therefore, it is not easy to produce the newest dataset consistently. Furthermore, a significant water shortage occurred in the 1970~1980s, and we assume that water shortage induced a complexity-oriented agricultural development. Thus, our primary contribution is to test the concept and measurement "complexity," a more extended study period and complete production/region types are more important than a new but lacking enough crop types dataset. Our approach is not perfect, but proper to test our assumption as a balanced result. We would like to see other latest and sufficient datasets developed and applied by our complexity-based method in the future, but that is not our primary goal.

*3. Line 75: It is recommended to use more concrete and precise description to introduce the meaning of $Mpc = 0$, rather than 'otherwise'.*

**Response:** Thanks for the recommendation, and sorry about the intuitiveness and readability of the mathematical description. Now, there is a more fluent and precise method introduction. We changed the above sentence like this:

[**P5, L93-L94**] Then, for $y \in [1978, 2008]$, we constructed such bipartite network $RCA_{ij}$ year by year to capture topology changes of Chinese VWF by time. Then, the network matrix $M$ is given by $M_{ij} = 1$ if $RCA_{ij} \geq 1$, and 0 if $RCA_{ij} < 1$.

*4. Figure 1: Is it possible to visually explain the Mathematical methods via the figures here? The description of methodology needs to be more intuitive.*

**Response:** Thanks for the advice! Since the method is a bit complicated, it takes us not a short time to design a better illustration. We rectified the original Figure 1 as this:

[Figure]

[*P4 a new figure*] **Figure 1:** Virtual water transfers between regions through multiple dominant crops. When the proportion of VWF from a region is large enough when considering a specific crop, we establish a connection between the region and the crop. In that way, we abstract region-crop bipartite networks for analysis. Here, we give a more straightforward illustration of our method: (1) Illustrating a nation comprised of three regions ($A$, $B$, and $C$) where three types of crops ($a$, $b$, and $c$) can be the productions for supporting virtual water flows (VWF). Region $B$ doesn't produce the crop $a$, and the region $C$ produce negligible crop $a$ and $b$. (2) When considering VWF volume, it is hard to compare the region $A$ and the $B$, as they both support 40% national total VWF. (3) However, when the structure is involved, their position in region-crop bipartite networks are different, typically in their differences of diversification ($N = 1$), uniqueness ($N = 2$) and competitiveness ($N = 3$) (the parameter N refers to levels of decomposition, detailed interpretation in section 2.4). (4) Therefore, considering both volume (by ignoring eligible VWF crops) and structure, the $GENEPY$ index is a method for distilling

information (see section 2.3). Our main results were derived from comparisons between indexes of YRB, China, and random networks (as a benchmark).

*5. Figure 4: Figure 4A shows that the diversification of YRB monotonically decreases during the whole time period. The uniqueness of YRB (indicated by Figure 4B, smaller number means higher uniqueness) has increased from 1978~1995 but then decreased during 1995~2000 before it increased again after 2000. This fluctuation has also been further verified by Figure 4C, is there any reason or background for this?*

**Response:** Thanks for the comment pointing such an interesting question. Firstly, there was some discussion in the previous manuscript:

> [**P10, L194-L199**] According to the literature Zhou, 2020, the slowdown of the growth of China's water consumption can be divided into two stages by the 1990s. While growth during the earlier period occurred mainly in arid and semi-arid areas (e.g., the YRB), growth during the latter period affected the whole country. Therefore, the increase of the national average level of complexity lagged behind that of the YRB. This lag was probably the result of the rest of China pursuing structural advantages later than the YRB.

Then, we did not analyze in-depth as the problem is too complicated to give a definite hypothesis. So here, we propose some hypotheses, as just a heuristic and to be proved:

(1) As mentioned above, higher complexity with decreasing VWF volume may hint at pursuing structural advantages under water shortage. Since 2000, water-saving transformations have been widespread all over the nation (according to (*1*)). However, because the network was constructed from a Relative Comparative Advantage (RCA, see method section), the Yellow River Basin lost relative comparative advantages from other regions' complexity developing, step by step.

(2) As one of the oldest farming culture origin, rich combinations of thermal and water supports potential ability in crop production of the YRB. Furthermore, irrigation gave the YRB more flexibility to pursue competitiveness because crops could get enough and timely water. However, water conservancy construction has weakened these relative advantages in other regions since the 2000s.

Summarized, we propose that the Relative Comparative Advantage (RCA) induced the fluctuation when advantages of YRB are no longer evident to the national average. However, a deeper reason why the relative comparative advantages changed is an open discussing question, and we just proposed our hypothesis in our manuscript.

*6. Is there any connection between virtual water network complexity and the resilience of water resources system against climate change and natural disasters in the target basin? This could be important especially when considering extreme weather events such as the extreme rain and flood events happened in northern China in summer 2021, which has caused both economy and agriculture damages. This might also show the importance of taking more recent time period into account and carrying out future climate scenario analysis in further steps of the research.*

**Response:** Thanks a lot for providing such an exciting question for open discussion.

The complexity of the VWF network means that the Yellow River basin's superior productivity and ability to provide water for crops throughout the country. On the one hand, if the water supply is not damaged, the Yellow River basin water network complexity (the ability to supply water for crops) provides enough tenacity for the river basin, has the ability of the Yellow River basin to meet with extreme weather may be better resilient than another area, because of a stronger relative comparing advantages. However, on the other hand, this complexity has brought robustness in crop production and trades of the Yellow River basin, which appears to produce other unaffected crops when some crops are affected by extreme weather.

However, it is unclear whether this supply force has imposed a burden on the local ecosystem as a whole and whether the increased complexity of the virtual water network means that there will be more artificial crops in the Yellow River basin that will affect the local ecological balance. With this in mind, will the Yellow River basin be more vulnerable when extreme weather strikes? What is more, when extreme weather strikes, especially drought or flood, it will lead to the loss of stable water supply capacity of the Yellow River Basin. In this case, the extreme weather will significantly threaten crops in the Yellow River basin and the crop circulation of the whole country because of embeddedness. Figuring this out will be necessary for how our agriculture responds to extreme weather in the future, and we hope to address this issue in future research.

**Bibliography**

Long, D. et al. South-to-North Water Diversion stabilizing Beijing's groundwater levels. *Nature Communication* 11, 3665 (2020).

F. Zhou, Y. Bo, P. Ciais, P. Dumas, Q. Tang, X. Wang, J. Liu, C. Zheng, J. Polcher, Z. Yin, M. Guimberteau, S. Peng, C. Ottle, X. Zhao, J. Zhao, Q. Tan, L. Chen, H. Shen, H.

Yang, S. Piao, H. Wang, Y. Wada, Deceleration of China's human water use and its key drivers. *Proceedings of the National Academy of Sciences*, 201909902 (2020).

---

## Author Response (AR2)

**Response letter**

Dear Editor in Chief,

Thank you very much and the anonymous reviewer for the valuable comments on our work again. We have followed the words, and that makes our manuscript clear. We hope this improved manuscript will satisfy you.

Thank you very much for your further attention and consideration. We are looking forward to hearing from you soon.

Best Regards

Yours Sincerely,

Shuang Song & Shuai Wang (on behalf of the author's team)
State Key Laboratory of Earth Surface Processes and Resource Ecology,
Faculty of Geographical Science,
Beijing Normal University
Beijing 100875, China
Tel: +86-10-58806579
Fax: +86-10-58806579
Email: songshgeo@mail.bnu.edu.cn

**Comments Response to reviewer 1**

General comment:

The quality of the manuscript has been significantly improved after the previous round of revision. The text description in the literature review has been modified and becomes more accurate. The methodology part can be understood easier with the new figure. The special fluctuation in the diversification trend has been explained. On the other hand, there are still some minor issues that need to be addressed. Therefore, a minor revision is recommended.

**Response:** Thank you for your help and recognition of our revision work!

Comment list:

*(1) P8-9: Figure 3 and Figure 4: The gap between 1985 and 2000 in Figures 4B and 4C (uniqueness and competitiveness) may correspond to the complexity gap between YRB and national in Figure 3A during the same period. However, the higher peak after 2000 in Figure 3A is not reflected in the uniqueness and competitiveness trend in Figures 4B and 4C. It is recommended to explain this in the article if there is a special reason for this result.*

**Response:** Thank you for pointing out such an interesting but previously overlooked point. As you remarked, we agree that adding some description would help the reader understand.

> *[P7, L150-L154]* Firstly, an indication of crop diversification ($N = 1$, Figure~A) was that almost every region was transferring virtual water through 7-8 dominant crops on average. The peak of diversification occurred before about 2000 when the major crop species supporting virtual water transfer were most abundant. Still, there is no significant difference between the YRB, China, and the random network (null models).

*(2) P6: Section 2.5 The introduction on 'Null models and sensitivity tests' is still not clear enough, especially the last sentence 'They were consistent with …' on P6. Also, the 'Random network' part of Figure 1. should be linked to Table 2. instead of simply using grey color.*

**Response:** Thank you for the valuable comment. Now, we polished figure 1 and linked it to Table 2, with some modifications in text for a better explanation:

> *[P6, L132-L137]* We randomly created provincial-crop bipartite networks for a sensitivity test. We calculated the same metrics as a comparable reference value to decide whether the influence of networks structure was trivial (Figure 1). The idea behind the randomization procedure is that we can create a null model starting from the data but shuffling the links of the network while conserving some of its statistical properties. We randomly generated (executed by Python 3.9 and Numpy 1.2) three scenarios where the bipartite networks have the same (1) number of edges, (2) edge sequences on provinces, and (3) edge sequences on crops, with the original dataset, respectively (Table 2 and Figure 1).

[Figure]

**Rectified Figure 1.** Virtual water transfers between regions through multiple dominant crops. When the proportion of VWF from a region is large enough when considering a specific crop, we establish a connection between the region and the crop. In that way, we abstract region-crop bipartite networks for analysis. Here, we give a more straightforward illustration of our method: (1) Illustrating a nation comprised of three regions (A, B, and C) where three types of crops (a, b, and c) can be the productions for supporting virtual water flows (VWF). Region B doesn't produce crop a, and region C produce a negligible crop a and b. (2) When considering VWF volume, it is hard to compare regions A and B, as they support 40% national total VWF. (3) However, when the structure is involved, their position in region-crop bipartite networks are different, typically in their differences of diversification (N = 1), uniqueness (N = 2) and competitiveness (N = 3) (the parameter N refers to levels of decomposition, detailed interpretation in section 2.4). (4) Therefore, considering both volume (by ignoring eligible VWF crops) and structure, the GENEPY index is a method for distilling information (see section 2.3). Our main results derive from comparisons between indexes of YRB, China, and random networks (as a benchmark). We generated three different null models by shuffling links in different ways and Table 2 gives them mathematical descriptions.